# Root mixing effects on belowground decomposition depend on mycorrhizal type

Lei Jiang[1], Stephan Hättenschwiler [2], Ning Ma [3,4], Jiajia Zheng[4,5], Wenhui Shi [1,5], Yeqing Ying[1], Shenggong Li[3,4], Han Yan[6,7] & Liang Kou [4,5] ✉

While there has been significant progress in understanding how species mixing affects leaf litter decomposition, the consequences for belowground root decomposition remains less known. This represents a critical knowledge gap, as roots are key contributors to soil carbon input. Here, we experimentally assess absorptive root decomposition in 138 paired-species combinations from 57 tree species, revealing significant non-additive mixing effects in 70% of all root combinations, with the majority of them decomposing faster than predicted from single species. Notably, non-additive effects occur only in mixtures containing at least one ectomycorrhizal species, with no net mixture effects in combinations of two arbuscular mycorrhizal species. We further find that these root mixing effects are associated with dissimilarities in condensed tannins across all mycorrhizal types and with nitrogen concentration when only ectomycorrhizal species are present. Overall, these root mixing effects are three times stronger than those documented for leaf litter decomposition in past studies. Collectively, our findings suggest that tree species mixing effects on decomposition are particularly robust belowground, especially in forests with ectomycorrhizal species of contrasting root chemistry. Absorptive root decomposition may have an essential role in how tree species mixing affects soil carbon and nutrient dynamics.

Plant and decomposer diversity can considerably modify litter decomposition[1–3], and may attenuate the negative effects of climate change on the decomposer system[4]. For example, mixtures of litter from different plant species often decompose at different rates compared to the predictions based on the individual component species decomposing singly[1,2,5]. However, theory and empirical studies have almost exclusively focused on aboveground leaf litter, and it is still largely unknown how root decomposition may be affected by diversity. This lack of knowledge is especially critical for the finest absorptive roots with high turnover rates, which play a particularly important role

in soil carbon (C) dynamics[6–8]. The few existing studies that addressed root mixing effects on decomposition included only a handful of different species and mixtures and found variable effects, including negative, positive, and purely additive effects[9–11]. The paucity of data on root mixing effects on decomposition and the absence of general patterns regarding a potential diversity-decomposition relationship belowground, hinder the general understanding of the importance of plant diversity in biogeochemical cycling of terrestrial ecosystems.

Physical and chemical litter characteristics are well-established predictor variables for decomposition[12,13]. The large majority of

[1]State Key Laboratory for Development and Utilization of Forest Food Resources, Zhejiang A&F University, Hangzhou, China. [2]CEFE, Univ Montpellier, CNRS, EPHE, IRD, Montpellier, France. [3]National Ecosystem Science Data Center, Key Laboratory of Ecosystem Network Observation and Modeling, Institute of Geographic Sciences and Natural Resources Research, Chinese Academy of Sciences, Beijing, China. [4]College of Resources and Environment, University of Chinese Academy of Sciences, Beijing, China. [5]Qianyanzhou Ecological Research Station, Key Laboratory of Ecosystem Network Observation and Modeling, Institute of Geographic Sciences and Natural Resources Research, Chinese Academy of Sciences, Beijing, China. [6]Freie Universität Berlin, Institut für Biologie, Berlin, Germany. [7]Berlin-Brandenburg Institute of Advanced Biodiversity Research, Berlin, Germany. ✉e-mail: koul@igsnrr.ac.cn

previous work established the relationships between litter characteristics, such as the concentrations of nitrogen (N) or lignin, and decomposition for leaf material. Some recent studies showed that these relationships may be driven by a different set of litter characteristics, such as the concentrations of condensed tannins or root morphological traits, for the finest absorptive roots compared to leaves[14,15]. Contrasting trait control over fine root compared to leaf litter decomposition may also have repercussions for the dynamics of root mixture decomposition, given that the mechanistic underpinnings of leaf litter mixture effects are largely based on chemical divergence among leaf litter from different plant species[3,5]. Indeed, a study[16] in a tropical rainforest on phosphorus (P) impoverished soils reported that differences in C, N and P concentrations between litter types, and the resulting stoichiometric dissimilarity (calculated with Rao's quadratic entropy[17]) in litter mixtures determined litter mixing effects on decomposition to a large extent. Stoichiometric dissimilarity as a driver of litter mixture effects, strongly support complementary resource availability to decomposers as an underlying mechanism[18]. However, as other studies showed, complementarity does not preclude the potential additional role of mass-ratio mechanisms, i.e. ecosystem processes are dominated by traits of the most abundant species[19,20], which is commonly assessed with community-weighted mean (CWM) trait values of litter mixtures[21]. While these trait-based mechanisms of litter mixture effects on decomposition are relatively well addressed for leaf litter, the current understanding for mixtures of roots from different plant species is extremely limited.

The absorptive roots of woody plants are predominantly associated with either arbuscular mycorrhizal (AM) or ectomycorrhizal (EcM) fungi[22]. Based on the mycorrhizal-associated nutrient economy framework[23], AM fungi directly scavenge inorganic nutrients released by saprotrophic microbes, and their hyphae readily colonize the upper mineral soil layer. In contrast, some EcM fungi apparently have access to organic nutrient sources, with their mycelium proliferating in organic soil horizons[23–26]. Furthermore, the root trait-based acquisition-defense-decomposition framework further proposes that AM and EcM tree species have contrasting belowground nutrient feedback loops[27]. The distinct mycorrhizal strategies and nutrient cycling modes of the two mycorrhizal types may infer contrasting chemical composition of root litter[26,28]. These potential dissimilarities in chemical traits may promote mechanisms related to resource use complementary by decomposers in mixed root litters[28]. As a consequence, stronger and more frequent synergistic effects can be expected in root mixtures composed of both mycorrhizal types compared to root mixtures with only one mycorrhizal type. However, this prediction remains untested as far as we know.

We address these questions in this study with absorptive roots from 57 mycorrhizal tree species associated with either AM or EcM fungi in the subtropical forests of China, contributing significantly to the global subtropical biodiversity[29]. We assess initial root traits potentially relevant for root decomposition, including morphological traits (diameter, specific root length, and root tissue density) and chemical traits (C, N, condensed tannins (CTs), and their ratios)[7,15,30] and run a microcosm experiment to quantify root decomposition rates of a total of 138 distinct two-species mixtures and their corresponding single species. To put these results in a larger context, we compare them to the reported decomposition of two-species woody leaf litter mixtures (n = 326) in the literature. We hypothesize that (i) there are more non-additive than additive mixing effects on root decomposition, and that positive non-additive (synergistic) effects are more frequent than negative non-additive (antagonistic) effects, similar to what is reported for leaf-litter, that (ii) positive non-additive mixture effects on decomposition increase with increasing root trait dissimilarity; and that (iii) trait dissimilarity and non-additive mixture effects on decomposition will be reinforced by mixing absorptive roots from

different mycorrhizal types, given the contrasting nutrient economies and the associated root traits between mycorrhizal types.

## Results

### Variation in root traits

Across the 57 tree species included in our experiment, we observed considerable variation in root morphological traits. Specific root length ranged from 7.8 m g$^{-1}$ in *Cryptomeria fortunei* to 215.8 m g$^{-1}$ in *Alniphyllum fortunei* with a coefficient of variation (CV) of 84.1% (Supplementary Tables 1 and 2). There was a 33-fold variation in root tissue density, ranging from 0.03 g cm$^{-3}$ in *Michelia macclurei* to 0.86 g cm$^{-3}$ in *Cryptomeria fortunei* (Supplementary Tables 1 and 2). Compared to specific root length and root tissue density, root diameter showed a smaller difference between the lowest and the highest values, of 0.32 mm and 0.62 mm, respectively, and with a CV of 18.3% (Supplementary Table 1). Root chemical traits showed the largest variation in the condensed tannins to nitrogen (CTs:N) ratio (CV = 57.7%), ranging from 0.5 in *Michelia figo* to 7.5 in *Cerasus serrulate*. The variation in root C (CV = 8.4%) was smaller than that in root N (CV = 24.3%). Total CTs ranged from 12.0 mg g$^{-1}$ in *Michelia figo* to 83.5 mg g$^{-1}$ in *Choerospondias axillaris* with a CV of 37.6% (Supplementary Tables 1 and 2). The extractable CTs accounted for 78% of total CTs and were correlated positively with total CTs (Supplementary Fig. 1).

We calculated the community-weighted mean (CWM) traits for the two-species mixtures, which were by definition within the smallest and largest values measured in individual species, thus covering a somewhat smaller range in trait values compared to the single species (Supplementary Tables 1–3). Accordingly, the variation in the average traits of the 138 different two-species mixtures ranged from 5.7% in root C to 54.6% in specific root length (Supplementary Table 3). For these 138 two-species mixtures, we also calculated the trait dissimilarity (absolute difference), with the largest variation observed for root diameter (CV = 108.8%) and the lowest variation observed for root N concentration (CV = 67.1%).

### Root mass loss

Root mass loss after 12 weeks of decomposition varied more than three-fold among the single species treatments, ranging from 9.0% in *Castanopsis carlesii* to 31.7% in *Michelia figo* (Supplementary Tables 1 and 2), with an average of 16.6 ± 5.6% (mean ± SD). Mass loss also varied considerably among the 138 two-species mixtures (Fig. 1a), and notably beyond the highest mass loss reported in the most rapidly decomposing litter of *Michelia figo* (see above). Across all two-species mixtures, mass loss ranged from 10.3% in *Castanopsis carlesii + Pinus massoniana* to 33.1% in *Liriodendron chinense + Michelia figo* with an average of 19.2 ± 5.0% and a CV of 25.8% (Supplementary Table 3).

With hierarchical partitioning analyses (HPA) we evaluated the relative contribution of each trait to the variation in root mass loss (Table 1). For single species, root diameter contributed the most, followed by root tissue density and C:N. This ranking changed for CWM trait values in two-species mixtures with C concentration contributing the most followed by root CTs, and CTs:N (Table 1). Overall, however, CWM traits varied less in their relative contribution to the decomposition of root mixtures compared to single species roots. We used linear analysis to explore the bivariate relationships between root trait and decomposition and found that mass loss was positively correlated with root diameter (F = 11.09, df = 1, r = 0.41, P = 0.002 for single species, F = 12.70, df = 1, r = 0.29, P < 0.001 for two-species mixtures), but negatively correlated with root CTs (F = 3.93, df = 1, r = −0.26, P = 0.052 for single species, F = 15.04, df = 1, r = −0.32, P < 0.001 for two-species mixtures), C:N (F = 8.83, df = 1, r = −0.37, P = 0.004 for single species, F = 13.84, df = 1, r = −0.30, P < 0.001 for two-species mixtures), and CTs:N (F = 6.42, df = 1, r = −0.32, P = 0.014 for single species, F = 13.98, df = 1, r = −0.31, P < 0.001 for two-species mixtures) for single species and for two-species mixtures as well (Table 1).

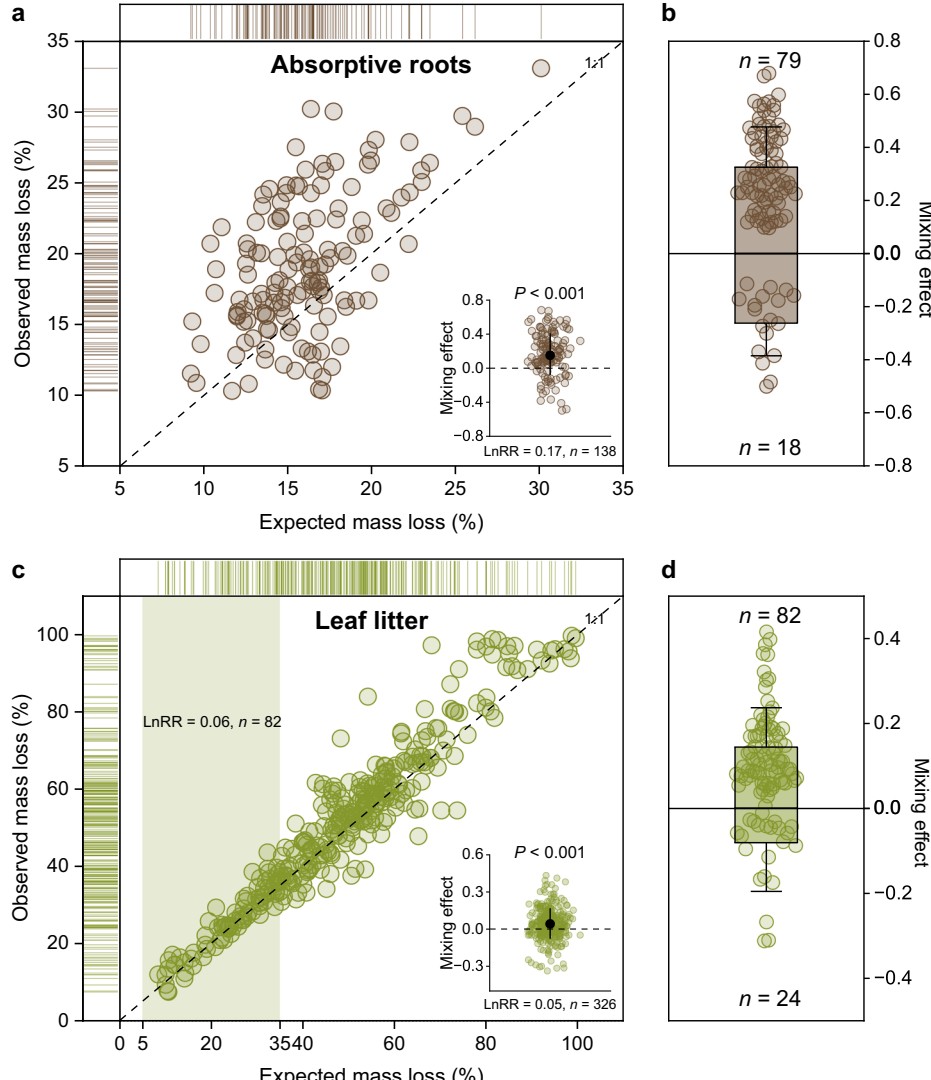

**Fig. 1 | The observed and expected mass loss and mixing effects for absorptive roots and leaf litter decomposition. a** Observed mass loss as a function of the expected mass loss for absorptive roots decomposing in microcosms of our laboratory experiment ($n = 138$). The brown dots above the 1:1 line indicate that the observed mass loss was greater than the expected mass loss. The insert at the right bottom shows the corresponding mixing effect (LnRR) of 138 paired-species root litter combinations, where the black point and line represent mean and standard deviation. The significance was based on two sided $t$ tests. **b** The combinations with significant synergistic effects ($n = 79$) above the zero line, and with significant antagonistic effects ($n = 18$) below the zero line from a total of 138 two-species root mixtures. Black points and lines represent means and standard deviations. **c** Observed mass loss as a function of the expected mass loss for two-species leaf

litter mixtures extracted from the literature ($n = 326$). The green dots above the 1:1 line indicate that the observed mass loss was greater than the expected mass loss. The light green shaded area represents the range of mass loss measured in our root decomposition experiment. The insert at the right bottom shows the corresponding mixing effect (LnRR) of 326 paired-species leaf litter combinations, where the black point and line represent mean and standard deviation. The significance was based on two sided $t$ tests. **d** The combinations with significant synergistic effects ($n = 82$) above the zero line, and with significant antagonistic effects ($n = 24$) below the zero line from the total of 326 two-species leaf litter mixtures. Black points and lines represent means and standard deviations. Source data are provided as a Source Data file.

## Mixing effects on root mass loss

In the majority of the 138 root mixtures, the observed mass loss was higher than the mass loss expected from the respective single species treatments (Fig. 1a). When calculated as the response ratio between observed and expected mass loss in litter mixtures (LnRR = ln ($ML_{obs}$/$ML_{exp}$)), we found a range from −0.50 in the root mixture of *Elaeocarpus sylvestris* + *Liquidambar formosana* to 0.68 in the root mixture of *Castanopsis carlesii* + *Carpinus viminea* with an overall mean value of 0.17 ± 0.24 ($t_{137} = 7.95$, $P < 0.001$, *Cohen's d* = 0.68, 95% confidence intervals from 0.12 to 0.21; Fig. 1a; Supplementary Table 3). A total of 70% ($n = 97$) of all root mixtures showed significant non-additive mixing effects. Synergistic effects emerged in more than half of the

mixtures (57%, $n = 79$), and 13% ($n = 18$) showed antagonistic effects (Fig. 1b).

The relative contribution of each CWM trait evaluated by HPA, showed that root diameter was the most important across all mixtures (49%, Supplementary Fig. 2a). Root diameter was negatively correlated with mixing effects ($F = 8.75$, $df = 1$, $r = −0.25$, $P = 0.004$; Supplementary Table 4), i.e. mixing effects were smaller with increasing CWM of root diameter. Based on trait dissimilarity, the highest-ranking trait for all combinations was also root diameter (27%), followed by root CTs (22%) and CTs:N (21%) (Fig. 2a). The mixing effects were negatively correlated with the dissimilarity in root diameter ($F = 4.63$, $df = 1$, $r = −0.18$, $P = 0.033$; Supplementary

**Table 1 | The relative contribution (%) of each of the eight root traits to the variation in root mass loss for single-species treatments and two-species mixtures (based on community-weighted mean (CWM) traits) using hierarchical partitioning analyses (HPA), and the relationships between CWM trait and mass loss for single-species and two-species mixture using linear regression**

| Trait | Relative contribution (%) | | Relationship | | | | | | | |
|---|---|---|---|---|---|---|---|---|---|---|
| | Single species (n = 57) | Two-species mixtures (n = 138) | Single species (n = 57) | | | | Two-species mixtures (n = 138) | | | |
| | | | F | df | r | P | F | df | r | P |
| Root diameter (RD, mm) | 31.5 | 12.3 | 11.09 | 1 | 0.41 | **0.002** | 12.70 | 1 | 0.29 | **<0.001** |
| Specific root length (SRL, m g⁻¹) | 5.9 | 3.2 | 0.28 | 1 | 0.07 | 0.598 | 0.14 | 1 | 0.03 | 0.711 |
| Root tissue density (RTD, g cm⁻³) | 20.8 | 10.8 | 4.78 | 1 | −0.28 | **0.033** | 2.04 | 1 | −0.12 | 0.156 |
| Root carbon concentration (C, mg g⁻¹) | 1.8 | 17.8 | 0.22 | 1 | −0.06 | 0.641 | 4.38 | 1 | −0.18 | **0.038** |
| Root nitrogen concentration (N, mg g⁻¹) | 8.5 | 12.4 | 5.36 | 1 | 0.30 | **0.024** | 10.71 | 1 | 0.27 | **0.001** |
| Total condensed tannins (CTs, mg g⁻¹) | 5.4 | 16.1 | 3.93 | 1 | −0.26 | 0.052 | 15.04 | 1 | −0.32 | **<0.001** |
| Carbon to nitrogen ratio (C:N) | 16.6 | 12.5 | 8.83 | 1 | −0.37 | **0.004** | 13.84 | 1 | −0.30 | **<0.001** |
| Condensed tannins to nitrogen ratio (CTs:N) | 9.6 | 14.9 | 6.42 | 1 | −0.32 | **0.014** | 13.98 | 1 | −0.31 | **<0.001** |

Ratios are based on mass. The statistical significance of the linear regression was assessed using two-sided $F$-tests. Significant values are in bold. Source data are provided as a Source Data file.

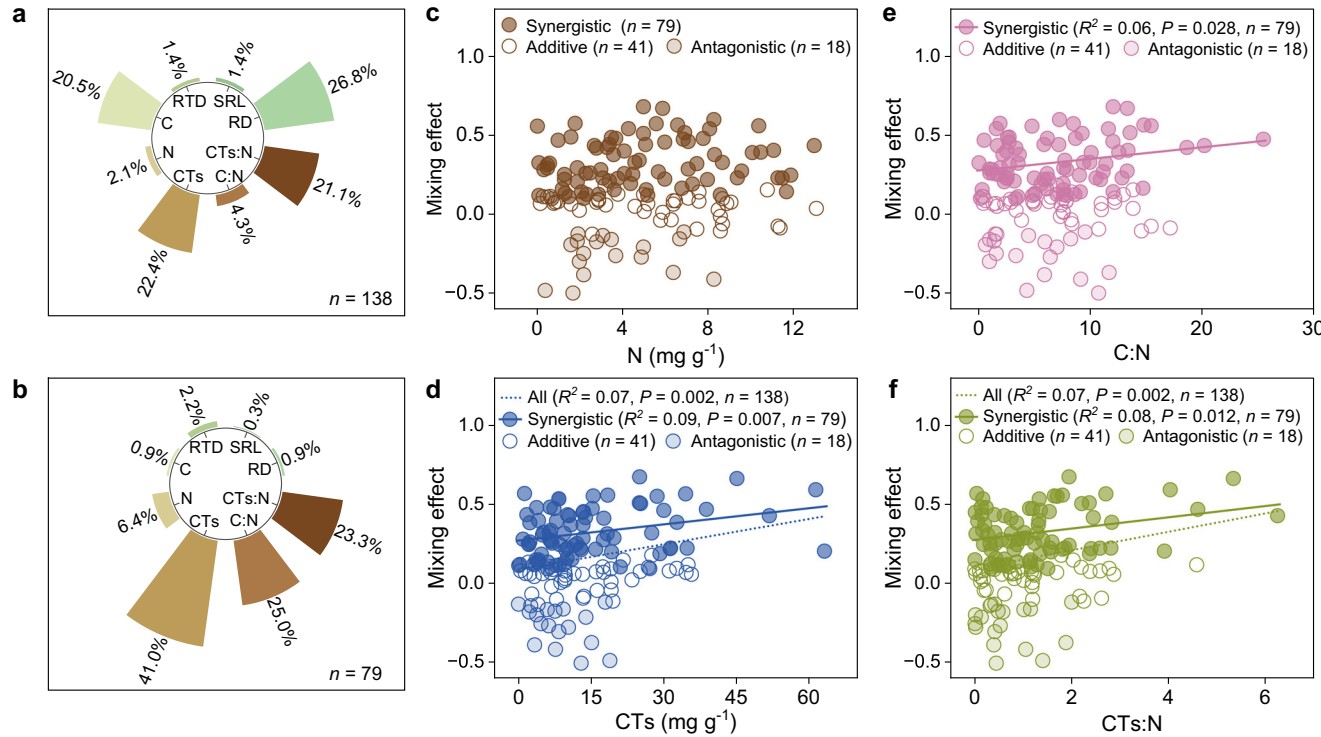

**Fig. 2 | Mixing effects on root decomposition as a function of trait dissimilarity. a** The relative contribution (%) of trait dissimilarity (expressed as the absolute difference between the two component species in the mixture) on mixing effects of all root combinations ($n = 138$) based on hierarchical partitioning analyses. **b** The relative contribution (%) of trait dissimilarity on mixing effects of root combinations with statistically significant synergistic effects ($n = 79$ out of 138 combinations in total) based on hierarchical partitioning analyses. **c**–**f** Mixing effects on root decomposition as a function of trait dissimilarity: N (**c**), CTs (**d**), C:N (**e**), and CTs:N (**f**). Open circles are for mixtures with purely additive effects, dark closed circles are for significant positive (synergistic) mixing effects, and light closed circles are for significant negative (antagonistic) mixing effects. The linear regressions are either fitted across all data (all 138 mixtures, dotted lines) or across only the mixtures with synergistic effects (a total of 79 mixtures, solid lines). There were no significant correlations for mixtures with antagonistic effects (Supplementary Table 4). The statistical significance of the linear regression was assessed using two-sided $F$-tests (Supplementary Table 4). RD, root diameter (mm); SRL, specific root length (m g⁻¹); RTD, root tissue density (g cm⁻³); C, root carbon concentration (mg g⁻¹); N, root nitrogen concentration (mg g); CTs, total condensed tannins concentration (mg g⁻¹); C:N, root carbon to nitrogen ratio; CTs:N, root condensed tannins to nitrogen ratio; Ratios are based on mass. Source data are provided as a Source Data file.

Table 4), i.e. the larger the difference in root diameter between the two species in the mixture the smaller the mixing effects, but positively correlated with dissimilarity in root CTs ($F = 9.95$, $df = 1$, $r = 0.26$, $P = 0.002$) and CTs:N ($F = 10.08$, $df = 1$, $r = 0.26$, $P = 0.002$; Supplementary Table 4, Fig. 2d, f). With the attempt to explore potential underlying mechanisms for mixing effects, we also analyzed the mixtures showing statistically significant synergistic effects ($n = 79$) separately. The highest-ranking CWM trait for synergistic combinations only was root CTs, followed by root N and CTs:N (Supplementary Fig. 2b). However, none of the relationships between CWM traits and synergistic mixing effects were significant (Supplementary Table 4). The highest-ranking dissimilarity trait for synergistic combinations only, was root CTs, followed by root C:N and CTs:N (Fig. 2b), with all of these three dissimilarity traits showing significant positive correlations in CTs ($F = 7.72$, $df = 1$, $r = 0.30$, $P = 0.007$), C:N ($F = 5.00$, $df = 1$, $r = 0.25$, $P = 0.028$) and CTs:N ($F = 6.69$, $df = 1$, $r = 0.28$, $P = 0.012$; Fig. 2d–f) for only the synergistic mixtures. However, the predictive power of these correlations was weak with only 10% or less of total variation in mixing effects accounted for. Similarly, when we analyzed exclusively the mixtures showing statistically significant antagonistic effects ($n = 18$), the importance of CWM traits decreased in the order of root diameter, root CTs:N, and CTs (Supplementary Fig. 2c). Despite the low number of only 18 mixtures showing antagonistic effects the negative correlation between root diameter and antagonistic mixing effects persisted ($F = 5.16$, $df = 1$, $r = -0.49$, $P = 0.037$; Supplementary Table 4). The importance of trait dissimilarity for only the mixtures with antagonistic effects decreased in the order of root C:N, root diameter, and root CTs (Supplementary Fig. 2d), but there were no significant correlations between dissimilarity traits and antagonistic mixing effects (Supplementary Table 4).

In order to provide context with the more abundant literature on leaf litter mixing effects on decomposition than what is presently available for absorptive root decomposition, we ran the same analysis for a total of 326 two-species leaf litter mixtures of woody species extracted from the literature. The overall mean mixing effect on leaf litter decomposition ($0.05 \pm 0.12$, $n = 326$) was weaker than that on root decomposition by using sampling of an equivalent number of observations with replacement ($P < 0.001$; Fig. 1). For 222 out of the total of 326 two-species mixtures, the difference between observed and expected mass loss was explicitly tested in these previous studies. For these 222 two-species mixtures, 48% showed non-additive effects ($n = 116$) with 37% being synergistic ($n = 82$) and 11% being antagonistic ($n = 24$; Fig. 1d). We also distinguished the data based on experimental conditions, i.e. field experiments and microcosm experiments in the laboratory (Supplementary Fig. 3). The field experiments showed stronger synergistic effects on mixture decomposition ($0.05 \pm 0.12$, $n = 266$; $P < 0.001$) than the laboratory experiments ($0.02 \pm 0.12$, $n = 60$; $P > 0.05$).

## The effects of mycorrhizal types

Across all 57 tree species, root diameter was significantly higher in AM than in EcM associated tree species, but no other root trait we measured differed between the two mycorrhizal groups (Supplementary Tables 5 and 6). When mixing roots from two species with AM association, the mean mixing effect was not significantly different from zero, indicating purely additive mixture effects on mass loss overall ($t_{56} = 1.970$, $P = 0.055$, Cohen's $d = 0.267$, 95% confidence intervals from $-0.002$ to $0.151$; Fig. 3a). However, when AM species were mixed with a second species associated with EcM, or when two species with EcM association were mixed, the mean mixing effect on root mass loss was overall positive, indicating synergistic mixture effects ($t_{46} = 7.200$, $P < 0.001$, Cohen's $d = 1.050$, 95% confidence intervals from $0.028$ to $0.147$ for AM/EcM mixtures; $t_{36} = 7.330$, $P < 0.001$, Cohen's $d = 0.1.205$, 95% confidence intervals from $0.180$ to $0.318$ for pure EcM mixtures; Fig. 3a). The mean response ratio (LnRR) was 0.07, 0.25, and 0.20 for

pure AM mixtures, pure EcM mixtures, and mixed AM/EcM mixtures, respectively (Fig. 3a). In addition to the highest LnRR overall, mixtures with only EcM-species had the highest proportion of synergistic effects (28 out of 37 EcM mixtures which represents 76%), with only 5% (2 mixtures) showing significant antagonistic effects. When species of either mycorrhizal type were mixed, the proportion of mixtures with synergistic effects was 68% (representing 32 out of 47 AM/EcM mixtures). Only four mixtures showed significant antagonistic effects (9% of all mixtures). A lower proportion of 57% of all mixtures with only AM associated species showed non-additive effects compared to the other two types of mixtures (81% for EcM only mixtures, and 77% for AM/EcM mixtures), and they were relatively equally divided into synergistic (19 out of a total of 54 AM mixtures) and antagonistic (12 mixtures) effects (Fig. 3a, b). When considering the synergistic combinations only, however, the mean mixing effect did not differ among the three mycorrhizal-type groups ($P > 0.05$; Fig. 3b; Supplementary Table 7).

We also used HPA for the assessment of the relative importance of the different root traits in the prediction of mycorrhizal type-specific mixing effects based on trait dissimilarity. For pure AM mixtures, root CTs and CTs:N dissimilarity contributed the most when all AM mixtures were considered ($n = 54$), while root diameter, CTs, and specific root length were the main predictors for synergistic effects ($n = 19$) (Fig. 3c). When roots from two species of either one of the two mycorrhizal types were mixed, root CTs and CTs:N dissimilarity were the main predictors across all mixtures ($n = 47$), and root CTs:N, CTs, and diameter were the main predictors for mixtures with synergistic mixing effects only ($n = 32$) (Fig. 3c). Finally, for pure EcM mixtures, root N and C:N dissimilarity were the most important predictors when all mixtures were included ($n = 37$), and root N alone contributed more than 50% to synergistic mixing effects, with C:N as the second most important trait ($n = 28$) (Fig. 3c).

Linear regression analyses showed that root CTs and CTs:N dissimilarity were positively correlated with mixing effects across all AM mixtures ($F = 6.31$, $df = 1$, $r = 0.33$, $P = 0.015$ for CTs; $F = 5.36$, $df = 1$, $r = 0.31$, $P = 0.025$ for CTs:N; Fig. 3e, g, Supplementary Table 8). However, these were not significant when only the mixtures with synergistic mixing effects were included (Supplementary Table 8). In only the AM mixtures showing antagonistic effects, root C dissimilarity contributed the most the mixing effects, showing also significant correlation with mixing effects ($F = 7.52$, $df = 1$, $r = -0.66$, $P = 0.021$; Supplementary Table 8). For AM/EcM mixtures, root CTs:N dissimilarity was positively correlated with mixing effects across all mixtures ($F = 4.90$, $df = 1$, $r = 0.31$, $P = 0.032$; Fig. 3g), but not when only the mixtures with synergistic effects were included in the test (Supplementary Table 9). For pure EcM mixtures, dissimilarities in N concentration and in C:N were positively correlated with mixing effects of both groups of points, all mixtures ($F = 10.08$, $df = 1$, $r = 0.47$, $P = 0.003$ for N; $F = 19.51$ $df = 1$, $r = 0.60$, $P < 0.001$ for C:N) or only the mixtures with synergistic effects ($F = 17.89$, $df = 1$, $r = 0.64$, $P < 0.001$ for N; $F = 10.84$, $df = 1$, $r = 0.54$, $P = 0.003$ for C:N; Fig. 3d, f; Supplementary Table 10).

## Discussion

The importance of plant roots and their mycorrhizal partners in driving soil C storage and ecosystem C and nutrient dynamics are widely recognized[6,8,31], but the role of plant diversity in root decomposition has been mostly neglected so far. Our large root mixture experiment including 138 two-species mixtures from a total of 57 tree species, showed that decomposition indeed changes when absorptive roots from two species decompose together. More than two thirds of all mixtures showed significant non-additive mixture effects and 81% of those were synergistic, that is mixtures decomposed faster than what is predicted based on single-species decomposition. To put these results in context, we ran in parallel the same analysis for the much better studied leaf litter decomposition with data taken from the

literature (Fig. 1). This comparison showed that the frequency of significant mixture effects (70% in our study compared to 48% for leaf litter mixtures) as well as the response ratio (LnRR of 0.17 in our study compared to 0.05 for leaf litter mixtures) are substantially larger for absorptive roots than for leaf litter. This indicates that litter mixture effects may be more pronounced belowground than aboveground suggesting a particularly important role of species mixing in litter decomposition and related biogeochemical cycling within the soil.

However, our relatively short-term experiment under controlled laboratory conditions may be difficult to compare to the majority of field studies over longer time periods for leaf litter mixture decomposition. When analyzing the experiments with leaf litter mixtures done under controlled laboratory conditions separately from those in the field (60 compared to 266 leaf litter mixtures, respectively; Supplementary Fig. 3), we observed that mixture effects were actually smaller in the lab studies than in the field. A likely reason for smaller mixture effects in the laboratory may be the notoriously absent soil fauna, a part of the decomposer community that can have a particularly strong contribution to mixture effects[5,16,32]. This suggests that mixture effects may be rather underestimated in the lab and that the effects we reported here may even be greater under field conditions in presence of the entire decomposer community, in particular soil fauna that was absent in our experiment. Whether this is really the case remains to be shown in future field studies.

A global synthesis of diversity-decomposition relationships for leaf litter showed that litter mixture effects emerge most strongly in the early stage of decomposition (before 40% of litter mass loss)[2]. Therefore, the stronger responses in root than leaf litter mixtures may also be related to the shorter duration of our experiment with root decomposition still in its initial stage compared to the generally longer duration of leaf litter experiments reaching later stages of decomposition (mean root mass loss was 19% in our study compared to 51% in the leaf litter studies). However, when we restricted the comparison to the same range of mass loss for leaf litter, the response ratio was still considerably lower for leaf litter mixture effects (0.06; Fig. 1c) than for absorptive root mixtures. There is also no apparent biome bias because when we analyzed the leaf-litter mixtures of only subtropical species, similar to our root mixture study, the response ratio remained similarly low as well (0.05; Supplementary Fig. 4). Collectively, these lines of evidence suggest that the reported stronger root than leaf litter mixture effects may represent a general pattern, which however, needs confirmation from other forest biomes and covering later stages of decomposition.

The high proportion of synergistic mixture effects is consistent with our first hypothesis stating that similar to previous reports on leaf litter mixing effects[2,33], root mixtures exhibit non-additive effects on decomposition with overall rather synergistic than antagonistic effects. The stimulation of decomposition in response to species mixing is quite substantial with 26% higher mass loss than predicted for the 79 mixtures with significant synergistic effects. The strong synergistic effect on root mixture decomposition suggests that tree species mixing can facilitate both above- and below-ground litter decomposition. It has been reported that absorptive roots that often have high levels of chemical defenses decompose quite slowly[15,30,34,35]. Our findings imply that belowground C and nutrient cycling in species-rich forests may be faster compared to forests dominated by a single species. According to the Microbial Efficiency-Matrix Stabilization framework, faster decomposition may foster more microbial products which further facilitates the formation of stable mineral-associated organic matter through more direct contact and easier chemical binding of microbial products to mineral surfaces[36]. Therefore, faster root mixture decomposition may also enhance soil C stabilization, which we did not explicitly test in our study, and which would need to be addressed in future studies preferably under field conditions.

The rate of decomposition is commonly regulated by a number of physical and chemical characteristics of the litter material[13,32,37]. In our study, two morphological traits, root diameter (RD) and root tissue density (RTD) contributed more than 52% to the variation in mass loss of single tree species roots (Table 1). In contrast, the three CWM traits contributing the most to the variation in root mass loss of mixtures were chemical traits (i.e. the concentrations of C and CTs and CTs:N ratio), collectively accounting for 48.8% of the variation. These differences suggest that the trait-decomposition relationship is modified when roots from two tree species are mixed compared to single-species root litter. Compared to a previous study that reported tannins as a major determinant for the decomposition of absorptive roots, leading to decreasing decomposition rates with increasing tannin concentrations[15], our data confirmed the potential role of CTs only for the decomposition of root mixtures, but not for roots from single tree species. Because Sun et al.'s study[15] covered a much longer time period of root decomposition with more advanced decomposition stages than in our study, the results may not be directly comparable as trait-decomposition relationships have been shown to change over time[38,39]. The retarding effect of tannins on decomposition may be explained by their protein binding property, or as more recently suggested by forming recalcitrant compounds by binding to fungal cell wall-derived chitin[40,41]. This latter mechanism may be particularly important for the highly mycorrhizal absorptive roots, especially for AM species with a more intimate intracellular contact between plant and fungal tissues.

The relative importance of CWM traits (mass ratio hypothesis) and trait dissimilarity (resource complementarity hypothesis) has been addressed for leaf litter mixture decomposition[21], but not for root mixture decomposition. We found that the mixing effects on root decomposition is mostly explained by trait dissimilarity rather than CWM traits in contrast to acting conjointly on leaf litter mixture decomposition[21], at least based on the traits we measured in our study (Fig. 2; Supplementary Fig. 2; Supplementary Table 4). Trait dissimilarity rather than CWM trait driven mixture effects is in support of our second hypothesis, but the predictive power was small with a maximum of 10% of the variation in mixture effects explained across all mycorrhizal types (see Fig. 2; Supplementary Table 4). Although predictions were better when analyzed within mycorrhizal groups separately as we discuss below, none of our models accounted for more than 41% of the variation in mixture effects. We may have missed some additional root traits in our analysis, such as lignin, bound phenolics, non-structural carbohydrates or nutrients other than N, which add to trait divergence in roots[27] and may contribute to regulate decomposition[15]. It would be interesting to evaluate the robustness of the relative role of root traits in driving root mixture decomposition in future studies including a wider spectrum of root traits. A recurrent difficulty in root decomposition work is the use of live root tissues as it is excessively difficult to extract naturally senesced root litter from the soil, especially for a large number of species as in our case. Some of the reported root trait values could therefore differ in true root litter, in particular those influenced by resorption processes such as nutrients. This would likely influence decomposition, as well as the reported trait dissimilarity among species. Our data may thus, underestimate trait dissimilarity and differences in decomposition because differences in resorption efficiency during senescence is expected to increase interspecific differences in nutrient-related traits[42–44], although potentially less so in roots than in leaves with lower resorption in roots compared to leaves[45].

Leaf litter from AM and EcM plants generally exhibits distinct nutrient contents, lignin to N ratios, and decomposition rates[23,46]. Some of these differences documented for leaf litter are also observed in absorptive roots, although studies measuring chemical and physical characteristics in absorptive roots are still few[15,27,30]. The comparatively few available data on absorptive root traits showed a somewhat less consistent trait difference between AM and EcM species. While root

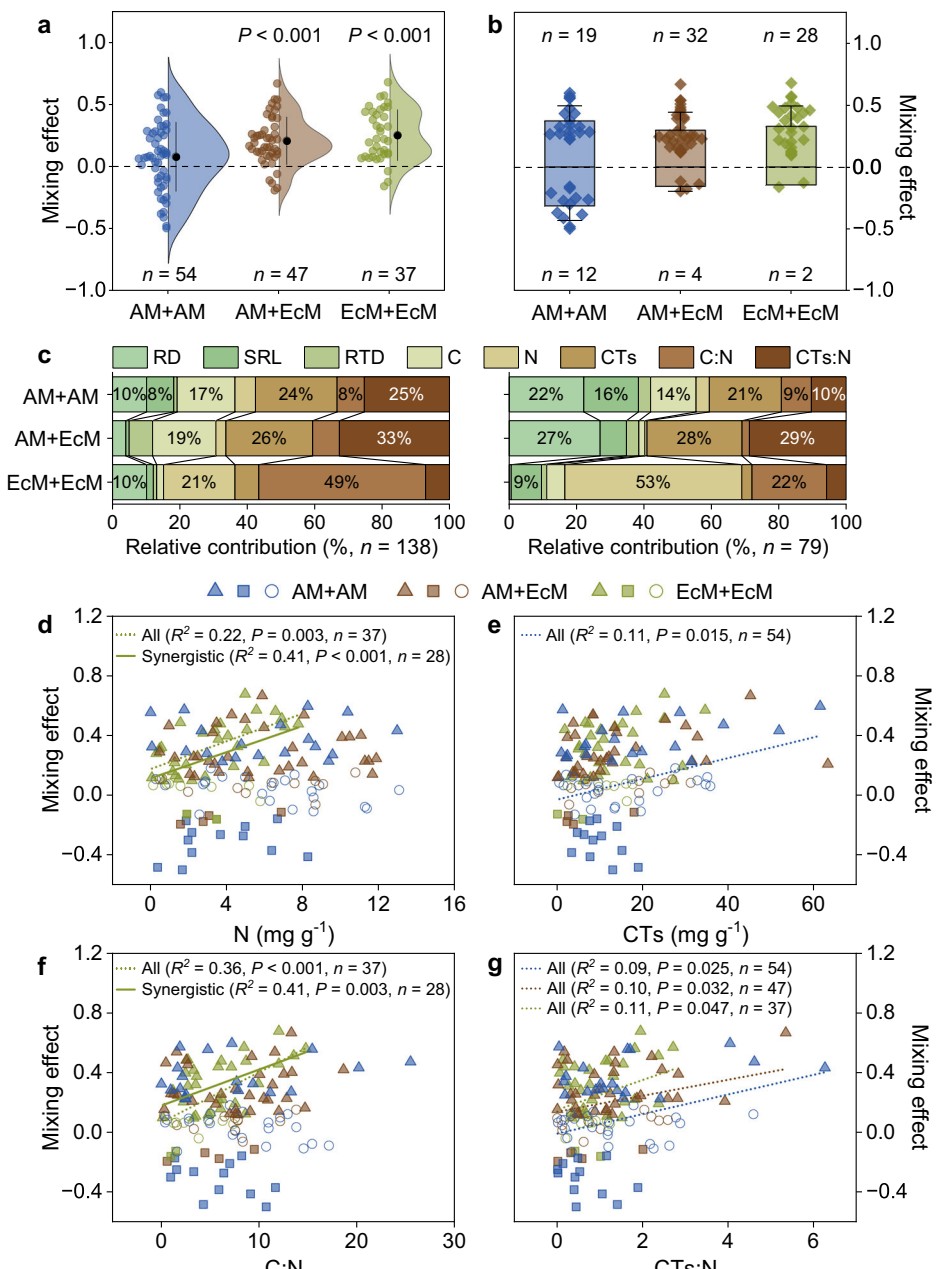

**Fig. 3 | Mycorrhizal type-specific root mixing effects on decomposition as a function of trait dissimilarity. a** Mycorrhizal type-specific mixing effects (AM+AM mixtures in blue, $n = 54$; AM+EcM mixtures in brown, $n = 47$; and EcM+EcM mixtures in green, $n = 37$). The black points and lines represent the means and standard deviations. The significance was reported for mixing effects tested using two sided $t$-tests. **b** The mixing effects of only the significant non-additive mixing effects (synergistic – above zero and antagonistic – below zero). Black points and lines represent means and standard deviations. **c** Hierarchical partitioning analyses to determine the relative contribution (%) of trait dissimilarity (expressed as the absolute difference between the two component species in the mixture) on mixing effects of all root combinations (on the left) or of only the root combinations showing synergistic effects (on the right) for the three mycorrhizal type combinations. **d**–**g** Mixing effects on root decomposition as a function of trait dissimilarity: N (**c**), CTs (**d**), C:N (**e**), and CTs:N (**f**). Open circles are for mixtures with

purely additive effects, solid triangles are for significant positive (synergistic) mixing effects, and solid squares are for significant negative (antagonistic) mixing effects. The linear regressions are either fitted across all data (all AM+AM mixtures, $n = 54$, blue dotted lines; all AM+EcM mixtures, $n = 47$, brown dotted lines; all EcM+EcM mixtures, $n = 37$, green dotted lines) or across only the mixtures with synergistic effects (AM+AM mixtures, $n = 19$, blue solid lines; AM+EcM mixtures, $n = 32$, brown solid lines; EcM+EcM mixtures, $n = 28$, green solid lines). The statistical significance of the linear regression was assessed using two sided $F$-tests (Supplementary Tables 8, 9, 10). AM arbuscular mycorrhizae, EcM ectomycorrhizae, RD root diameter (mm), SRL specific root length (m $g^{-1}$), RTD, root tissue density (g $cm^{-3}$), C root carbon concentration (mg $g^{-1}$), N root nitrogen concentration (mg $g^{-1}$), CTs root total condensed tannins concentration (mg $g^{-1}$), C:N root carbon to nitrogen ratio, CTs:N root condensed tannins to nitrogen ratio; Ratios are based on mass. Source data are provided as a Source Data file.

diameters are generally larger in AM than EcM species[15,27,30], N concentrations are higher and tannin and lignin concentrations lower in absorptive roots of AM than EcM species in some[27,30], but not in other studies[15]. Based on these trait differences between the two mycorrhizal types, we expected that the contrasting mycorrhizal types would be a

strong determinant of non-additive mixing effects. In contrast to our hypothesis, non-additivity of mixture effects did not differ in root combinations of both mycorrhizal types from those with identical mycorrhizal types (Fig. 3b). The differences in root characteristics between AM and EcM species were less strong in the set of species

studied here (Supplementary Tables 5 and 6) compared to some of the previous studies, perhaps because of the high within mycorrhizal type variability of the species included. In fact, the divergence in CTs and CTs:N that correlated with root mixing effects, showed no significant differences among the three groups of mycorrhizal-type combinations (Supplementary Fig. 5).

Instead, we found an unexpected clear effect of the presence of an EcM species in the root mixture. Synergistic mixture effects were more frequent in mixtures of AM and EcM species and in mixtures with two EcM species than in mixtures with two AM species, with no significant net mixture effect across all mixtures in the latter (Fig. 3a, b). These results suggest that the presence of an EcM species within the root mixture facilitates synergistic mixture effects, and it seems that this is increasingly associated to the dissimilarity in N concentration with increasing EcM dominance in the mixture (Fig. 3c) instead of CTs and CTs:N ratio observed across all synergistic mixtures regardless of mycorrhizal type. In fact, hierarchical partitioning analyses indicated that dissimilarity in N concentration accounted for more than half of the variation in synergistic effects of root mixtures composed of only EcM species in strong contrast to the negligible N effect in mixtures containing at least one AM species (Fig. 3c). Mixtures including AM species on the other hand showed an important contribution of CTs and CTs:N ratio as well as of root diameter to the synergistic mixing effects. This mycorrhizal-type specific trait contribution may suggest different mechanisms driving synergistic litter mixing, which may be related to the different nature of mycorrhizal symbiosis in the two groups, with an extracellular type of symbiosis in EcM species and an intracellular type of symbiosis in AM species[47]. These distinct ways of how the fungal tissues are associated to the root tissue could promote tannin interactions with plant-derived proteins and/or fungal chitin in AM roots more than in EcM roots. Synergistic effects with EcM associated roots, in contrast may be similar to those observed in leaf litter mixtures, where N-based complementary resource use or N transfer may be an important mechanism[3,48]. There may be additional morphological and/or physiological differences between roots of the two types of symbiosis, which may explain the distinct role of EcM species within root mixtures, but which were not quantified here. If the particularly strong effect of roots from plants associated to EcM in root mixture decomposition is confirmed in future studies, it would be interesting to further elucidate the underlying mechanisms of the observed differences between AM and EcM associated roots in mixtures.

We conclude that decomposition of absorptive roots is overall faster when roots from two tree species are mixed compared to single species decomposition and that this mixture effect is stronger than that documented for mixed leaf litter decomposition. This indicates that species mixing effects on decomposition may be particularly important belowground. The stimulating effect on decomposition by root mixing was mostly driven by roots from trees associated with EcM fungi, suggesting that the presence of EcM tree species in mixed forests may have a particularly strong impact on belowground decomposition dynamics. Although we identified root trait dissimilarity as a potential mechanism for mixture effects on decomposition, this needs confirmation in future studies including a greater number of root traits and an assessment of later stages in decomposition. Given the potential ecosystem-level consequences in C and nutrient dynamics following changes in decomposition, species mixing effects during decomposition of absorptive roots need more consideration in future studies to better account for tree species mixing effects on soil processes and ecosystem functioning.

## Methods

### Sampling sites and tree species

We used root material from 57 tree species sampled at two sites of subtropical forests in Jiangxi Province of Southeastern China. The selected tree species belonged to 44 genera from 27 families, which encompassed a large range in successional status, canopy position, maximum height and age, leaf habit and longevity, and mycorrhizal status. More details of species descriptions are provided in Supplementary Table 1. The first sampling site was at the Qianyanzhou Ecological Station (26°44′39″N, 115°03′33″E, 102 m. a.s.l.) with the zonal vegetation restored from wastelands around 1985 by initiating the establishment of forests by planting Slash pine (*P. elliottii*), Masson pine (*Pinus massoniana*), Chinese fir (*Cunninghamia lanceolata*), and a mix of different broad-leaved tree species[49]. The sampling included a total of 13 tree species from that first site, all relatively abundant and dominant in the present forest canopy. In order to increase the number of tree species included in the experiment, we collected an additional 44 species from a second site with a species-rich subtropical forest located at the Yangming Mountain Nature Reserve (25°39′26″N, 114°18′49″E, 400 m. a.s.l.)[47]. Both sites are characterized by similar climatic conditions, typical for the subtropical climate zone, with mean annual temperatures and precipitations of 17.9 °C and 1475 mm at the first site[47,50] and 17.7 °C and 1587 mm at the second site[26,47]. Soil characteristics at the two sites are also largely comparable, having weathered from red sandstone and mudstone, and classified as Inceptisol, according to China soil taxonomy and USDA soil taxonomy[51,52].

### Field sampling and sample processing

For each species, roots were collected in the field from three to five healthy and mature trees with a similar diameter at breast height and with a distance of at least 10 m between individual trees. We determined four 1 m × 1 m plots, one in each cardinal direction, around each target individual tree at the same distance for an individual tree, but ranging between 1.5 m and 2.5 m from the tree stem among individual trees. In each plot, we excavated roots from the surface soil (0–20 cm) and traced them back to the stem to verify their identity[53]. Then, the intact fine roots including at least the first five root orders were carefully cut from the main lateral roots. All root samples were sealed into valve bags and transported to the laboratory typically within two hours. We collected the surface soil (0–20 cm) within the root sampling plots for all individuals of the 57 tree species to represent the sampling location from the two forest sites. In the laboratory, all soil samples were sieved through a 2 mm mesh and then homogenized for each site of collection. Then, we weighed the same amount of 100 kg from both collection sites and mixed them together for one single homogeneous substrate used in the root decomposition experiment. The soil properties of this single substrate are given in Supplementary Table 11.

Before dissection, intact root samples were cleaned with deionized water to exclude attached soil following the procedure described by Guo[54]. Roots were then dissected according to the branching-order classification method[55], keeping only the finest first- and second-order roots for our experiment. We will refer to these two first root orders as "absorptive roots" in contrast to the first three orders proposed previously for absorptive roots[56]. We did this to reduce heterogeneity among species, because some tree species may vary strongly in the shape and diameter of the third-order roots[54]. Then, absorptive roots of individual trees were mixed for each species. We took a representative subsample of fresh roots that was immediately stored at −20 °C for morphological trait measurements. The remaining root material was oven-dried (60 °C, 48 h) to constant weight for chemical trait measurements and the decomposition experiment.

### Root trait and soil measurements

The frozen root subsamples were spread out on glass panes to avoid any overlap among roots and scanned on an Epson Expression 10000 XL scanner (Seiko Epson Corporation, Suwa, Nagano, Japan) at a resolution of 400 dpi[27]. Root diameter, length, and volume were measured using WinRHIZO Arabido version 2012b (Regents

Instruments Inc., Quebec City, Canada). All scanned samples were oven-dried to constant weight (root dry mass) to calculate specific root length (root length/root dry mass) and root tissue density (root dry mass/root volume).

A representative subsample from the oven-dried root material was ground using a Retsch MM 400 Mixer Mill (Retsch GmnH, Haan, Germany) for subsequent chemical analysis. Approximately 60–80 mg milled root samples were wrapped with tin foil and their C and N concentrations were determined by a Vario Macro cube elemental analyser (Elementar Analysensysteme GmbH, Langenselbold, Germany). The CTs concentration was determined by the acid butanol method. Briefly, 0.05 g of root sample was extracted with acetone (70%) in a sonicator (30 min) and centrifuged (10 min, 4500 rpm, 2264× $g$) three times. We added ascorbic acid (0.015 g) to the supernatant to measure all the extracted CTs and the precipitate was freeze-dried and used to measure CTs on the unextracted fraction. Total CTs concentration was the sum of extracted and unextracted CTs[57–59].

Soil total C and N concentrations were determined from air-dried subsamples of the bulk soil used for microcosm construction (we worked with four laboratory replications), following the same method and using the same elemental analyzer as mentioned above for root samples. The inorganic N (nitrate and ammonium) was extracted from the same bulk soil, but on fresh soil subsamples before air-drying (2 mol L$^{-1}$ KCl, 50 ml), and then analyzed with a continuous-flow autoanalyzer (Autoanalyzer 3; Bran and Luebbe, Norderstedt, Germany), we again worked with four laboratory replications. Available P was extracted with HCl (0.025 mol L$^{-1}$)-NH$_4$F (0.03 mol L$^{-1}$) from air-dried soil subsamples (four laboratory replications) and its concentration was determined colourimetrically using an ascorbic acid molybdate analysis on the continuous-flow autoanalyzer[60]. Soil pH was measured in a soil suspension with a soil to water ratio of 1:2.5 (weight/volume) using a digital pH meter (Mettler Toledo, Greifensee, Switzerland) using the same subsamples of air-dried bulk soil used for mineral P analyses. More methodological details are given in Jiang et al.[61] and Zheng et al.[27] and soil characteristics are given in Supplementary Table 11.

## Root decomposition experiment

Given the large range of tree species included in our test, we decided to run a laboratory experiment under controlled conditions to clearly disentangle root mixture effects and trait control on decomposition from a multitude of environmental factors that would vary in the field. We constructed a microcosm experiment using polyvinyl chloride jars (10 cm length, 8 cm diameter) to quantify the decomposition of absorptive roots (1st- and 2nd-order roots) of individual tree species and their mixtures. Each jar was filled with fresh soil (100.0 ± 0.003 g) from the homogenized bulk soil (see above). Other than sieving through a 2 mm mesh for homogenization and removing larger debris like stones or pieces of wood, the soil and the microbial and microfauna communities remained unaltered. Other components of the naturally occurring decomposer organisms from the mesofauna and the macrofauna were partially or completely removed by sieving (particularly the macrofauna except for very small juveniles or eggs that were potentially present in the sieved soil). We embedded one nylon litterbag (7 cm × 7 cm, 0.1 mm mesh width) filled with a total of 0.5 ± 0.0002 g root material (equal amounts of 0.25 ± 0.0002 g of each species for two-species mixtures) horizontally into each microcosm at 1 cm soil depth. We had a total of 195 litter treatments, including 57 single-species and 138 two-species mixtures and a total of 585 microcosms (three replicates per litter treatment).

We determined species combinations of two-species litter mixtures based on root diameter and mycorrhizal type, which was identified as a strong predictor of absorptive root decomposition[27,30]. To do so, we first ranked the 57 species included in the study according to

their root diameter and calculated the quantiles to establish the thresholds for dividing the data into three equal segments, the lower third is categorized as "small" (root diameter between 0.32 mm and 0.35 mm, 23 species) the middle third as "intermediate" (root diameter between 0.36 mm and 0.39 mm, 15 species) and the upper third as "large" (root diameter between 0.40 mm and 0.62 mm, 19 species). We obtained three types of combinations by mixing species from the three different diameter classes. The first type was the combination of a species from the class with a large root diameter with a species from the class with a small root diameter. The second type was the combination of a species from the class with either a large or a small root diameter with a species from the intermediate root diameter class. The third type was the combination of two species from the same class of root diameter. We further classified these tree species into 46 AM and 11 EcM species, following their assignments of symbiotic guilds based on Yan et al.[47] and Soudzilovskaia et al.[62]. We determined the species pairs randomly and if happened to be a species without enough root material we replaced it by another species from the same class and particular attention was paid to the balance of sample sizes for different mycorrhizal-type groups. This resulted in roots of some species being used multiple times in combinations while others were used only once. Overall, we obtained 138 mixtures including 54 pure AM mixtures, 47 mixed AM/EcM mixtures, and 37 pure EcM mixtures. The microcosms were incubated in the dark at 25 ± 0.02 °C and we added distilled water to maintain soil humidity at 60% of water-holding capacity every two weeks by weighing each microcosm throughout the experiment. Each microcosm was well covered with a perforated adherent film that minimized humidity losses but permitted gaseous exchange.

Litterbags in microcosms were harvested after 12 weeks of laboratory incubation. Soil that adhered to the litterbags was carefully brushed off. The remaining root samples in the litter bag were oven-dried (60 °C) to constant weight to determine the mass loss. The percentage observed mass loss $ML_{obs}$ after 12 weeks of incubation was calculated as:

$$\text{ML}_{obs} = (mass_b - mass_a)/mass_b \times 100 \tag{1}$$

where $mass_b$ and $mass_a$ are the dry weight (g) of the root material before and after incubation, respectively. The expected mass loss $ML_{exp}$ of the root mixture treatment was calculated as the mean mass loss of the component species decomposing alone[10,63],

$$\text{ML}_{exp} = (ML_1 + ML_2)/2 \tag{2}$$

where $ML_1$ and $ML_2$ are the respective mass loss percentage of species 1 and species 2. The mixing effect on root decomposition was calculated as the response ratio of the differences between observed and expected mass loss,

$$LnRR = \ln\left(ML_{obs}/ML_{exp}\right) \tag{3}$$

The mixing effect was synergistic or antagonistic when $LnRR$ was significantly higher or lower than zero, respectively. Additive effects occurred when $LnRR$ was not significantly different from zero.

## Calculation of root traits dissimilarity and community-weighted mean traits

To explore which root trait predict litter mixture decomposition best, we calculated two metrics of litter mixtures for each root trait independently. Specifically, we used the absolute difference to calculate root trait dissimilarity in mixtures[63]:

$$\text{absolute difference} = \left| X_{ik} - X_{jk} \right| \tag{4}$$

where $X_{ik}$ and $X_{jk}$ represent the values of trait $k$ in species $i$ and species $j$, respectively. This absolute difference was calculated for each root trait value individually to explore the relationships between root traits and mixing effects.

We also used community-weighted mean (CWM) values of root traits to explore the importance of average trait values in root mass loss of single species and mixtures, when the root mass of two species was even in litter mixtures[21], following:

$$CWM_j = \sum_{i=1}^{n} p_{ij} \times t_{ij} \qquad (5)$$

where $p_{ij}$ is the relative abundance of the species $i$ in the litter mixture $j$, and $t_{ij}$ is the mean trait value of the species $i$ in the litter mixture $j$. In our study, $n$ is the number of species in a mixture, i.e. $n = 2$.

### Assembly of the leaf-litter mixture decomposition data

To be able to put the mixing effects on absorptive-root mixtures in a larger context of the currently available literature on litter mixing effects, we compared it to leaf litter mixing because data on root mixing are still extremely scarce. To do so, we extracted the data of equal-ratio mixtures of leaf litter for paired species from previous reviews on leaf litter diversity effects on decomposition in forest ecosystems[2,64]. We further complemented these data with a search of peer-reviewed articles published before October 2024 using the ISI Web of Science, Google Scholar, and China National Knowledge Infrastructure. The studies had to meet the following criteria: (i) they focused on the decomposition of leaf litter mixtures from woody plant species in forest ecosystems; (ii) the mixtures consisted of exactly two species with equal ratios; (iii) the decomposition experiments used either litterbags in field conditions or microcosm in fully controlled laboratory conditions; and (iv) the studies reported mass loss, mass remaining, or decomposition constants of mixtures of leaf litter and their corresponding single litter species treatments. Overall, a total of 326 paired-species combinations in 86 published papers met these criteria and were included here. Therefore, 222 combinations explicitly reported the difference between the observed and expected mass loss, that is, significant synergistic, antagonistic, or additive effects, while the other 104 combinations did not.

### Data analysis

All data were tested for normality and log$_{10}$-transformed when necessary. Outliers defined as data with an absolute value of Z-score greater than three were excluded to eliminate the effects of extreme values on leaf-litter mixture decomposition. One sample $t$-test was used to test whether the mixing effect (LnRR) on absorptive root or leaf-litter decomposition differs significantly from zero. We used hierarchical partitioning analyses (HPA) to calculate the relative contribution of each trait to the mass loss of absorptive roots for single species and two-species mixtures based on the CWM metric. We further used linear regressions to explore the bivariate relationships between key traits and mass loss of absorptive roots for single species and two-species mixtures based on the CWM metric. Similar to the approach including all data describe above, we further used HPA to calculate the relative contribution of each trait based on CWM or trait dissimilarity to the mixing effect in all mixtures and the significantly non-additive mixtures. We then determined the bivariate relationships between root trait dissimilarity or CWM and mixing effects on root decomposition separately for all mixtures, for the mixtures with synergistic effects, and for mixtures with antagonistic effects using linear regressions. We separately analyzed the two groups because the traits driving the potential underlying mechanisms are expected to differ between positive and negative mixture effects.

In a series of further analyses, we explored the effects of mycorrhizal types on root trait dissimilarities and mixing effects on root decomposition. Because of unequal numbers of AM (46 species) and EcM (11 species) associations of the sampled tree species, we applied a bootstrap method by random sampling with an equivalent number of observations with replacement[65,66] to test how many times differences in initial root traits and mass loss were significant between the two mycorrhizal types. Similarly, because the root mixtures in our experiment represented different numbers of mycorrhizal type combinations (54 mixtures included two AM species, 37 mixtures included two EcM species, and 47 mixtures included one of each mycorrhizal type), we constructed bootstrap methodology by random sampling with an equivalent number of observations with replacement to test how the mixing of the same or different mycorrhizal types modulated the mixing effects. We ran the same test also for the statistically significant synergistic combinations only. We further conducted a sensitivity analysis of the established bootstrap method with different replacements (iteration numbers = 500, 1000, 2000, 5000), and metrics showing relative variation ((max - min)/min) <10% were stable. We used the HPA to calculate the relative contribution of each trait to the mixing effect in all mixtures and the significantly non-additive mixtures from different mycorrhizal-type groups. We then determined the bivariate relationships between root trait dissimilarity and mixing effects on root decomposition separately for all mixtures, for the mixtures with synergistic effects, and for mixtures with antagonistic effects from different mycorrhizal-type groups using linear regressions. All analyses were conducted in R version 4.3.2 and all figures were plotted in OriginPro (2024, Origin Lab., Hampton, MA, USA).

### Reporting summary

Further information on research design is available in the Nature Portfolio Reporting Summary linked to this article.

## Data availability

The data generated in this study have been deposited in the Figshare repository: https://doi.org/10.6084/m9.figshare.29151101. Source data are provided with this paper.

## Code availability

The R scripts used for the analysis in this study are openly available at Figshare: https://doi.org/10.6084/m9.figshare.29151101.

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

## Acknowledgements

This research is financially supported by grants from the National Key Research and Development Program of China (No. 2022YFD2201500 to W.S.) and the National Natural Science Foundation of China (Nos. 32222059 to L.K.; 32101306 to L.J.).

## Author contributions

L.K. and L.J. conceived the ideas and designed the study. L.J. collected and analyzed the data. S.H. offered thoughts on the analysis and graphics. N.M. and L.J. drew the graphics. L.K., L.J. and S.H. led the writing of the manuscript. J.Z., W.S., Y.Y., S.L. and H.Y. contributed substantially to the revisions of the draft.

## Competing interests

The authors declare no competing interests.
