## [Transparent Peer Review file · Nature Communications]

Root mixing effects on belowground decomposition depend on mycorrhizal type

Corresponding Author: Professor Liang Kou

Version 0:

Reviewer comments:

Reviewer #1

(Remarks to the Author)

Overall Comments

This is an ambitious project investigating fine root decomposition across many tree species and species pairs. The design of the study, methodology, and analysis all seem solid and well thought through. The study clearly advances our understanding of decomposition dynamics and the role of mixtures of litter types. The main advance is that it focuses on root decomposition which is little studied compared to leaf litter decomposition. The comparison of AM vs ECM species is also noteworthy and interesting. However, I feel the authors overinterpret their data in order to attempt inferences to the effects of biodiversity broadly on decomposition processes, nutrient cycling and stable soil C formation. In addition, I feel that there is an overselling of the synergistic effects of mixtures on decomposition and of the importance of “trait dissimilarity” in driving root decomposition. Overall, I suggest: 1) a more narrow focus on decomposition dynamics, 2) a more critical assessment of synergistic effects and trait dissimilarity as a driving factor in determining non-additive mixing effects.

Overinterpretation

In the abstract and the discussion the authors consistently overinterpret their findings in terms of “plant diversity” or “biodiversity” on root decomposition. In fact, their study simply contrasted single-species root decomposition with paired species root decomposition. An experimental design with multiple levels of species richness would be needed as has been done many times for investigations of the role of species diversity on ecosystem processes (e.g. the long running Cedar Creek study or reference #3 in this manuscript). In addition, the authors spend quite a bit of the discussion, conclusions and abstract extrapolating well beyond their data to draw inferences to species diversity effects on nutrient cycling and soil carbon stabilization. This study did not even examine nutrient dynamics in decomposing litter and only reports mass loss in the very early stages of decomposition. Therefore, I think it is inappropriate to attempt to draw any conclusions regarding nutrient cycling rates or soil organic carbon stabilization.

Synergistic Effects and Trait Dissimilarity

In the discussion (lines 313-322) the authors describe the “dominance of synergistic effects” and argue that “the strong synergistic effect on root mixture decomposition provides compelling evidence...” This seems to me to greatly overstate the importance of synergistic mixture effects from their data since only a bit more than half of the mixtures showed synergistic effects with increased decomposition. Thirty percent had no effect and 13% actually showed the opposite – decreased decomposition rate in mixtures. I don’t see this as synergistic effects being dominant at all. The conclusions and mechanistic discussion around “trait dissimilarities” as the main drivers of mixture effects also seems to be over sold in the results and discussion. First of all, I question the restriction of the data set to only the 57% of mixtures that had significant synergistic effects. What is the justification for doing this? If you really want to draw conclusions about driving mechanisms of mixture effects why would you truncate your data set nearly in half and exclude those with purely additive and antagonistic non-additive interactions? This seems highly questionable to me. But then, even with restricting the data set to those with significant synergistic effects the evidence for trait dissimilarities as an important driver seems exceedingly weak to me. In Figure 2 panels b, c, and d the R² values are all below 0.1, even in the restricted data set. In Figure 3 panels d-g, where these are broken down by mycorrhizal type we see some somewhat compelling relationships for dissimilarities in N and C:N, but only for a further truncated component to the data (ECM-ECM pairs). Across the entire study, trait dissimilarities do not appear to have any predictive power for understanding effects of root mixtures on early stage decomposition.

Additional Detailed Comments

L49 “Biodiversity” is too broad a term here to introduce your study. I think “plant species richness” would be more appropriate
L66-71. This section is quite vague, can you revise identifying the litter characteristics that are important for leaf and root decay?
L79-80 do you mean “resource complementarity” rather than “niche complementarity?” Wouldn’t niche complementarity need to refer to complementarity in niches of different decomposers; whereas, I think you are focused on the complementarity of resources in litter among different tree species.
L79-82. This section is extremely vague and jargony for anyone not working in this specific subfield of ecology. If you are going to use these terms would be good to provide explanations of what they mean
L96-100. The way this section is written it is unclear if someone else has posed this hypothesis previously or if you are posing it here for the first time.
L112-115. I think you would benefit from breaking Hypothesis 1 into two. As stated it is overly complicated
L168-172. This section needs a reference to a figure or table.
194-195 As described above, these correlations may be statistically significant but the relationships are exceedingly weak with almost no predictive power. It’s hard for me to believe that they have any ecological significance
L248-255. I believe that for all of these parameters in the HPA you are referring to community-weighted means and not dissimilarities, but it is unclear. Please make sure to state this clearly in this section.
L266 change “was’ to “has been”
L278-281 haven’t studied plant diversity per se, just single species vs paired species
L309-311 Delete this last sentence. You have no evidence to support such a broad conclusion
L318-333 this section is full of interpretations that go far beyond anything that you have data on. I think this needs to be deleted
L336-338. I don’t see the data in Table 1 supporting this conclusion 2 out of 3 morphological traits were significantly related to decomposition and the contributed the top two relative contributions in single species treatments.
L341-344. Again I find this conclusion confusing because it appears inconsistent with data reported in Table 1. Isn’t root diameter the best predictor in mixtures according to Table 1? Or at least comparable to some of the chemical factors?
L355 again I feel that you should be phrasing this as resource complementarity rather than niche complementarity
L428-440 These conclusions are extremely speculative and all are far, far beyond any data presented in this manuscript.
L481 Weighed not weighted
L547 Is this a typo did you really weigh out 100.002 and not 100.000?
L547-553. This all seems irrelevant since you enclosed roots in 0.1 mm opening mesh – none of these organisms can access them
L661-663. Here you say you did the bivariate relationships for synergistic and antagonistic combinations; however in Figs 2 and 3 you only report for significant synergistic. See my general comments above. It seems to me that you should not truncate your data set in this way, or at least need to provide a compelling justification for doing so
Figure 3. The ordering of parameters in panel C the does not make sense. I suggest ordering them as you did for Table 1: physical/morphological, then chemical absolute, then chemical ratios.
Table 1. capitalize Root tissue density to be consistent with other parameters.

Reviewer #2

(Remarks to the Author)

This study provides a novel exploration of how mycorrhizal type (AM and EcM) mediates diversity effects on belowground root decomposition. By systematically analyzing 138 root mixtures across 57 subtropical trees and integrating global dataset of leaf-litter decomposition, the authors convincingly demonstrate that absorptive-root mixtures exhibit stronger synergistic effects than leaf-litter mixtures, with ectomycorrhizal (EcM)-associated roots playing a central role. This work addresses a critical gap in assessing biodiversity effects on belowground root decomposition at a large scale. The experimental design, statistical rigor, and novel hypotheses align with Nature Communications’ standards, providing valuable insights and have broad implications for understanding soil carbon dynamics and ecosystem functioning. I have some specific suggestions to temper the study’s broader applicability.

First, the study provides valuable insights into early-stage decomposition processes during the 12-week incubation period. However, the focus on short-term dynamics leaves a critical gap in understanding long-term carbon (C) stabilization mechanisms, such as microbial necromass accumulation or mineral-associated organic matter formation. These processes are essential for predicting the persistence of soil organic carbon and its role in climate change mitigation. I suggest addressing the following point: the 12-week incubation period is suitable for capturing early-stage decomposition but fails to account for long-term C stabilization pathways. Including a discussion on how the observed early-stage decomposition might influence long-term C stabilization, even if these processes were not directly measured.

Second, the reliance on live roots, without accounting for pre-senescence nutrient resorption, risks misrepresenting decomposition processes in natural settings. Although a recent study reported that temperate root resorption rates (20-25%) are lower than foliar rates (55%) and there were no significant differences in nutrient resorption metrics of root between mycorrhizal types (Wang et al., 2025, *New Phytologist*, Nutrient resorption of leaves and roots coordinates with root nutrient acquisition strategies in a temperate forest), the generalizability of these values to subtropical tree species remains unclear. Furthermore, dead roots often undergo lignification during senescence, which live roots may not approximate. So, I suggest the authors should make it more explicit in the discussion section that the decomposition experiments based on live root tissue may differ from those using naturally senesced roots to further clarify whether the experimental approach may conservatively reflect in situ decomposition dynamics.

Third, while the focus on condensed tannins, nitrogen, and their ratio provides valuable insights into decomposition dynamics, a more comprehensive analysis integrating lignin content and non-structural carbohydrates would significantly enhance the mechanistic interpretation of results. Lignin, as a dominant recalcitrant compound, directly constrains microbial degradation through physicochemical barriers, while non-structural carbohydrates (e.g., soluble sugars, starch) can prime microbial activity, transiently accelerating early-stage decomposition even in tannin-rich tissues. To mitigate this, I suggest framing conclusions more cautiously and explicitly proposing future work to disentangle their roles.

Fourth, the manuscript specifies the use of 1000 bootstrap iterations for statistical analysis, which is a reasonable choice for balancing computational efficiency and precision. While 1000 iterations are commonly used in bootstrap analysis, the stability of the results should be explicitly verified, especially for readers who may question whether this number is sufficient for the specific dataset and analysis. Without sensitivity analysis, it is unclear whether the results would significantly change if a higher number of iterations were used. Conducting a sensitivity analysis by comparing results across different numbers of bootstrap iterations would involve recalculating key statistical metrics (e.g., means, standard deviations, confidence intervals) for each iteration count. Please include a supplementary figure or table summarizing the findings of the sensitivity analysis.

Fifth, please upload complete leaf litter dataset (species, locations, climate) to Dryad/Figshare, and provide R scripts for hierarchical partitioning analyses to ensure methodological transparency and reproducibility of the results.

Version 1:

Reviewer comments:

Reviewer #1

(Remarks to the Author)

I was Reviewer #1 on the original manuscript. I have read through the authors' responses to my original comments as well as reviewing the changes they made to manuscript text, figures and tables. I find that the authors gave thoughtful consideration to all of the issues raised in my original review. I think they did an excellent job in addressing my questions and concerns, and they provided clear and defensible explanations for their decisions. I very much appreciate the thoughtfulness and effort they put into this revision. I support acceptance of this manuscript.

Reviewer #2

(Remarks to the Author)

I'm glad that there is a great improvement in this version. My concerns have been settled properly and I think this ms is suitable for publication in Nature Communications.

Open Access This Peer Review File is licensed under a Creative Commons Attribution 4.0 International License, which permits use, sharing, adaptation, distribution and reproduction in any medium or format, as long as you give appropriate credit to the original author(s) and the source, provide a link to the Creative Commons license, and indicate if changes were

made.

Response to the reviewers' comments:

General remarks

We greatly appreciate the constructive and insightful comments by the two reviewers, which were very helpful for improving the manuscript, which we believe is now more balanced.

We have revised the manuscript according to these comments (highlighted in yellow in the manuscript).

Below we provide point-by-point responses (in blue font) to the reviewers' comments (in black font). We also add the text part that was modified in the manuscript in *blue italic*.

Point-by-point response to reviewers' comments

Reviewer #1 (Remarks to the Author):

Overall Comments

This is an ambitious project investigating fine root decomposition across many tree species and species pairs. The design of the study, methodology, and analysis all seem solid and well thought through. The study clearly advances our understanding of decomposition dynamics and the role of mixtures of litter types. The main advance is that it focuses on root decomposition which is little studied compared to leaf litter decomposition. The comparison of AM vs ECM species is also noteworthy and interesting. However, I feel the authors overinterpret their data in order to attempt inferences to the effects of biodiversity broadly on decomposition processes, nutrient cycling and stable soil C formation. In addition, I feel that there is an overselling of the synergistic effects of mixtures on decomposition and of the importance of "trait dissimilarity" in driving root decomposition. Overall, I suggest: 1) a more narrow focus on decomposition dynamics, 2) a more critical assessment of synergistic effects and trait dissimilarity as a driving factor in determining non-additive mixing effects.

Response: We thank the reviewer for this general appreciation and comments. We carefully considered the remarks on overselling of a part of our results, which we now reformulated more cautiously, including more critical discussion, in the revised version. Please see below for more details in how we incorporated the reviewer's concerns in the revised version.

Overinterpretation

In the abstract and the discussion the authors consistently overinterpret their findings in terms of "plant diversity" or "biodiversity" on root decomposition. In fact, their

study simply contrasted single-species root decomposition with paired species root decomposition. An experimental design with multiple levels of species richness would be needed as has been done many times for investigations of the role of species diversity on ecosystem processes (e.g. the long running Cedar Creek study or reference #3 in this manuscript). In addition, the authors spend quite a bit of the discussion, conclusions and abstract extrapolating well beyond their data to draw inferences to species diversity effects on nutrient cycling and soil carbon stabilization. This study did not even examine nutrient dynamics in decomposing litter and only reports mass loss in the very early stages of decomposition. Therefore, I think it is inappropriate to attempt to draw any conclusions regarding nutrient cycling rates or soil organic carbon stabilization.

Response: With a second look at our manuscript in the light of the reviewer's comments, we agree that part of the interpretations and conclusions extended too far beyond the collected data. Especially in the Abstract and the Conclusions, we avoided an extrapolation beyond what our data actually show. On the other hand, we find it important to provide some discussion how the changes in root decomposition may influence soil C and nutrient dynamics on the basis of the general knowledge on the relationships between decomposition and the dynamics in C and nutrients. This also allows to connect to the rich literature on the subject, which helps the reader to understand the potential implications of our findings. We acknowledge that the balance between an entirely factual discussion of just the results and what they may imply more generally is not easy to determine, but some of the latter is needed for an interesting and stimulating Discussion section. We tried to discuss potential implications cautiously and we explicitly emphasized the need for further investigation (for example on lines 344-347, 364-367, 480-486) for the confirmation of potential consequences of changed decomposition dynamics in mixed root litter.

We also agree that it is more appropriate to refer to species mixing instead of biodiversity when referring to the results of our study. We modified the manuscript accordingly avoiding generalizations that reach too far beyond of our experiment. In some parts of the Introduction and the Discussion when we refer to previous studies or in a more general context, we however, kept the terms 'plant diversity' or 'biodiversity' for a clear understanding.

Synergistic Effects and Trait Dissimilarity

In the discussion (lines 313-322) the authors describe the "dominance of synergistic effects" and argue that "the strong synergistic effect on root mixture decomposition provides compelling evidence..." This seems to me to greatly overstate the importance of synergistic mixture effects from their data since only a bit more than half of the mixtures showed synergistic affects with increased decomposition. Thirty percent had no effect and 13% actually showed the opposite – decreased decomposition rate in mixtures. I don't see this as synergistic effects being dominant at all. The conclusions and mechanistic discussion around "trait dissimilarities" as the

main drivers of mixture effects also seems to be over sold in the results and discussion. First of all, I question the restriction of the data set to only the 57% of mixtures that had significant synergistic effects. What is the justification for doing this? If you really want to draw conclusions about driving mechanisms of mixture effects why would you truncate your data set nearly in half and exclude those with purely additive and antagonistic non-additive interactions? This seems highly questionable to me. But then, even with restricting the data set to those with significant synergistic effects the evidence for trait dissimilarities as an important driver seems exceedingly weak to me. In Figure 2 panels b, c, and d the R² values are all below 0.1, even in the restricted data set. In Figure 3 panels d-g, where these are broken down by mycorrhizal type we see some somewhat compelling relationships for dissimilarities in N and C:N, but only for a further truncated component to the data (ECM-ECM pairs). Across the entire study, trait dissimilarities do not appear to have any predictive power for understanding effects of root mixtures on early stage decomposition.

Response: Thank you for your feedback on how we interpreted the mixture effects and the potential underlying mechanisms. We agree that in our attempt to provide a clear message, we tended to oversimplify the interpretation of the results in some parts of the manuscript. We now adopted a more moderate style to discuss and interpret our data, adding nuance and detail to the discussion. Some words were perhaps hastily chosen, like “dominance” to describe synergistic effects, which however, were significantly more frequent. We have a somewhat different view when exploring potential mechanisms for non-additive effects. We would argue that it makes little sense to look for mechanisms when the effect is absent. In other words, in an attempt to understand what could potentially drive non-additive mixture effects, such an evaluation needs to focus on mixtures that actually showed non-additive mixture effects. It is little convincing to extend such an analysis to mixtures with purely additive effects as there is no pattern to search for mechanisms. However, we modified the Figs. 2 and 3 by including also an analysis across the entire data set (all mixtures) in order to provide all the information to the reader with the comparison of the same relationships for only the mixtures with synergistic or antagonistic effects. We also added more explanation on these different relationships in the results by more clearly stating that in the case of non-additive mixture effects it may be relevant to understand by what root characteristics they are determined while acknowledging that non-additive mixture effects represent about two thirds of all mixtures tested.

We agree that even when analyzed only for the root mixtures with significant non-additive mixture effects, the correlations with trait dissimilarity are weak, and although significant, the low variability explained is not sufficient to pinpoint a clear mechanism. We substantially modified the text reporting and discussing these results to accommodate the reviewer’s remarks. Please see below for more details in the revised version.

Additional Detailed Comments

L49 “Biodiversity” is too broad a term here to introduce your study. I think “plant species richness” would be more appropriate.

Response: We agree that “biodiversity” is too broad and too vague when we refer more specifically to our study. Here in this first sentence, however, we intend to refer to the already existing literature on the topic, which includes various aspects of biodiversity. We think it is important to provide this general overview to a broad readership before focusing more specifically on the context of our study. We still modified the sentence to make it more explicit and less vague: “*Plant and decomposer diversity can considerably modify litter decomposition....*” (line 49).

L66-71. This section is quite vague, can you revise identifying the litter characteristics that are important for leaf and root decay?

Response: We revised this section accordingly by adding more explicit information on the litter traits that were reported to influence leaf and root decomposition. Please see lines 66-72.

“Physical and chemical litter characteristics are well-established predictor variables for decomposition^{12,13}. The large majority of previous work established the relationships between litter characteristics, such as the concentrations of nitrogen (N) or lignin, and decomposition for leaf material. Some recent studies showed that these relationships may be driven by a different set of litter characteristics, such as the concentrations of condensed tannins or root morphological traits, for the finest absorptive roots compared to leaves^{14,15}.”

L79-80 do you mean “resource complementarity” rather than “niche complementarity?” Wouldn’t niche complementarity need to refer to complementarity in niches of different decomposers; whereas, I think you are focused on the complementarity of resources in litter among different tree species.

Response: The two are connected, but we agree that it is clearer when using “resource complementarity” and revised accordingly: “*Stoichiometric dissimilarity as a driver of litter mixture effects, strongly support complementary resource availability to decomposers as an underlying mechanism¹⁸.”* (please see lines 81-83).

L79-82. This section is extremely vague and jargony for anyone not working in this specific subfield of ecology. If you are going to use these terms would be good to provide explanations of what they mean.

Response: We agree with the comment and modified the sentence accordingly: “*However, as other studies showed, complementarity does not preclude the potential additional role of mass-ratio mechanisms, i.e. ecosystem processes are dominated by traits of the most abundant species^{19,20}, which is typically assessed with community-weighted mean (CWM) trait values of litter mixtures²¹.”* (please see lines 83-87).

L96-100. The way this section is written it is unclear if someone else has posed this hypothesis previously or if you are posing it here for the first time.

Response: We modified the text to make it clearer that while distinct nutrient foraging of AM and EcM roots and associated differences in root chemistry have been studied and highlighted in previous studies, the hypothesis that mixing AM and EcM roots may stimulate decomposition remained untested (please see lines 98-108):

“Furthermore, the root trait-based acquisition-defense-decomposition framework further proposes that AM and EcM tree species have contrasting belowground nutrient feedback loops²⁷. The distinct mycorrhizal strategies and nutrient cycling modes of the two mycorrhizal types may infer contrasting chemical composition of root litter^{26,28}. These potential dissimilarities in chemical traits may promote mechanisms related to resource use complementary by decomposers in mixed root litters²⁸. As a consequence, stronger and more frequent synergistic effects can be expected in root mixtures composed of both mycorrhizal types compared to root mixtures with only one mycorrhizal type. However, this prediction remains untested as far as we know.”

L112-115. I think you would benefit from breaking Hypothesis 1 into two. As stated it is overly complicated

Response: We agree that the previous writing was a bit convoluted. We have now rewritten the hypothesis to present it more straightforwardly. However, we prefer to keep it as a single hypothesis, because the two parts are closely related. We now write (lines 120-128): *“We hypothesized that (i) there are more non-additive than additive mixing effects on root decomposition, and that positive non-additive (synergistic) effects are more frequent than negative non-additive (antagonistic) effects, similar to what is reported for leaf-litter; that (ii) positive non-additive mixture effects on decomposition increase with increasing root trait dissimilarity; and that (iii) trait dissimilarity and non-additive mixture effects on decomposition will be reinforced by mixing absorptive roots from different mycorrhizal types, given the contrasting nutrient economies and the associated root traits between mycorrhizal types.”*

L168-172. This section needs a reference to a figure or table.

Response: Yes, this was indeed missing in the previous version. We now added a reference to Table 1 (please see lines 174-181).

194-195 As described above, these correlations may be statistically significant but the relationships are exceedingly weak with almost no predictive power. It’s hard for me to believe that they have any ecological significance.

Response: As we stated above in the response to the general remarks, we agree that

this was overdone in the previous version. We also think that it is not really possible to firmly infer control by the divergence in specific root traits based on the presented data. There is some indication for how particular root traits may be involved in driving some of the non-additive mixture effects, especially when accounting for mycorrhizal type, but some other traits may have been missed with our selection of measured root traits. This clearly needs more investigation before drawing more firm conclusions. Accordingly, we modified the associated text passages throughout the manuscript. For example, we have reworded the sentence in the Abstract (line 39), results (lines 214-217) and explicitly acknowledged in the Discussion that the modest R^2 values warrant further investigation by incorporating additional root chemical traits (lines 399-421).

L248-255. I believe that for all of these parameters in the HPA you are referring to community-weighted means and not dissimilarities, but it is unclear. Please make sure to state this clearly in this section.

Response: Thank you for raising this issue. Actually, the parameters in the HPA focus on trait dissimilarity not community-weighted means. We have revised the text to make it clear. Please see lines 269-271.

“We also used HPA for the assessment of the relative importance of the different root traits in the prediction of mycorrhizal type-specific mixing effects based on trait dissimilarity.”

L266 change “was’ to “has been”

Response: We changed this accordingly (please see line 299).

L278-281 haven’t studied plant diversity per se, just single species vs paired species.

Response: Yes, we agree and following our response to the general remarks we replaced the terms of ‘diversity’ or ‘biodiversity’ throughout the manuscript (including in the title) by ‘richness’ or ‘mixing’, unless when we referring to the broader literature that looked at various aspects of biodiversity.

L309-311 Delete this last sentence. You have no evidence to support such a broad conclusion.

Response: We agree that this conclusion was too bold and we rephrased this sentence accordingly (please see lines 344-347).

“Collectively, these lines of evidence suggest that the reported stronger root than leaf litter mixture effects may represent a general pattern, which however, needs confirmation from other forest biomes and covering later stages of decomposition.”

L318-333 this section is full of interpretations that go far beyond anything that you have data on. I think this needs to be deleted

Response: We recognize that we stretched the interpretations of faster early-stage decomposition too much. We now revised this passage avoiding overstatements, while still putting the findings in a somewhat more general context of what a modified decomposition may imply. Please see lines 354-367:

“The strong synergistic effect on root mixture decomposition suggests that tree species mixing can facilitate both above- and below-ground litter decomposition. It has been reported that absorptive roots that often have high levels of chemical defenses decompose quite slowly^{15,30,34,35}. Our findings imply that belowground C and nutrient cycling in species-rich forests may be faster compared to forests dominated by a single species. According to the Microbial Efficiency-Matrix Stabilization framework, faster decomposition may foster more microbial products which further facilitates the formation of stable mineral-associated organic matter through more direct contact and easier chemical binding of microbial products to mineral surfaces³⁶. Therefore, faster root mixture decomposition may also enhance soil C stabilization, which we did not explicitly test in our study, and which would need to be addressed in future studies preferably under field conditions.”

L336-338. I don't see the data in Table 1 supporting this conclusion 2 out of 3 morphological traits were significantly related to decomposition and they contributed the top two relative contributions in single species treatments.

Response: Thank you for pointing out this incoherence. There was actually an error in Table 1 that we now corrected. We modified the text accordingly to (please see lines 370-376):

‘In our study, two morphological traits, root diameter (RD) and root tissue density (RTD) contributed more than 52% to the variation in mass loss of single tree species roots (Table 1). In contrast, the three CWM traits contributing the most to the variation in root mass loss of mixtures were chemical traits (i.e. the concentrations of C and CTs and CTs:N ratio), collectively accounting for 48.8% of the variation.’

L341-344. Again I find this conclusion confusing because it appears inconsistent with data reported in Table 1. Isn't root diameter the best predictor in mixtures according to Table 1? Or at least comparable to some of the chemical factors?

Response: Thank you for pointing this out. As noted in the answer to the previous query, an older version of Table 1 persisted in the manuscript, with the values for the mixing effects (LnRR) rather than mass loss, which we now corrected along with the text referring to this table. Please see response to L336-338 and lines 376-385.

“These differences suggest that the trait-decomposition relationship is modified when roots from two tree species are mixed compared to single-species root litter. Compared to a previous study that reported tannins as a major determinant for the decomposition of absorptive roots, leading to decreasing decomposition rates with

increasing tannin concentrations¹⁵, our data confirmed the potential role of CTs only for the decomposition of root mixtures, but not for roots from single tree species. Because Sun et al.'s study¹⁵ covered a much longer time period of root decomposition with more advanced decomposition stages than in our study, the results may not be directly comparable as trait-decomposition relationships have been shown to change over time^{38,39}.”

L355 again I feel that you should be phrasing this as resource complementarity rather than niche complementarity

Response: Yes, we revised accordingly (line 394).

L428-440 These conclusions are extremely speculative and all are far, far beyond any data presented in this manuscript.

Response: We revised the conclusions paragraph considerably in order to stick closer to the data we collected (lines 480-490). Please see also our response to the general comments above.

“The stimulating effect on decomposition by root mixing was mostly driven by roots from trees associated with EcM fungi, suggesting that the presence of EcM tree species in mixed forests may have a particularly strong impact on belowground decomposition dynamics. Although we identified root trait dissimilarity as a potential mechanism for mixture effects on decomposition, this needs confirmation in future studies including a greater number of root traits and an assessment of later stages in decomposition. Given the potential ecosystem-level consequences in C and nutrient dynamics following changes in decomposition, species mixing effects during decomposition of absorptive roots need more consideration in future studies to better account for tree species mixing effects on soil processes and ecosystem functioning.”

L481 Weighed not weighted

Response: We corrected as suggested (line 531).

L547 Is this a typo did you really weigh out 100.002 and not 100.000?

Response: We adjusted the decimals.

L547-553. This all seems irrelevant since you enclosed roots in 0.1 mm opening mesh – none of these organisms can access them

Response: Even without direct access to the root material, the presence of these organisms could have had some indirect effects adding to unaccounted variation in the data depending on differences in their abundance/identity among replicates. We kept the description of how we actually prepared the soil.

L661-663. Here you say you did the bivariate relationships for synergistic and antagonistic combinations; however in Figs 2 and 3 you only report for significant synergistic. See my general comments above. It seems to me that you should not truncate your data set in this way, or at least need to provide a compelling justification for doing so

Response: As we describe above in response to the general comments, we modified this analysis by running three parallel tests with 1) data of all mixtures, 2) data of the mixtures with synergistic effects, and 3) data of the mixtures with antagonistic effects. With the aim to search for potential trait-related mechanisms for mixture effects we believe it is important to evaluate synergistic and antagonistic mixtures separately and on their own (see above for further explanation and lines 195-207, 269-294, 709-716, 734-740).

Figure 3. The ordering of parameters in panel C the does not make sense. I suggest ordering them as you did for Table 1: physical/morphological, then chemical absolute, then chemical ratios.

Response: Following this suggestion, we reordered the eight parameters based on morphological, then chemical absolute, then chemical ratios in panel C. Please see the newly drawn Fig. 3c.

Table 1. capitalize Root tissue density to be consistent with other parameters.

Response: We modified this accordingly.

Reviewer #2 (Remarks to the Author):

This study provides a novel exploration of how mycorrhizal type (AM and EcM) mediates diversity effects on belowground root decomposition. By systematically analyzing 138 root mixtures across 57 subtropical trees and integrating global dataset of leaf-litter decomposition, the authors convincingly demonstrate that absorptive-root mixtures exhibit stronger synergistic effects than leaf-litter mixtures, with ectomycorrhizal (EcM)-associated roots playing a central role. This work addresses a critical gap in assessing biodiversity effects on belowground root decomposition at a large scale. The experimental design, statistical rigor, and novel hypotheses align with Nature Communications' standards, providing valuable insights and have broad implications for understanding soil carbon dynamics and ecosystem functioning. I have some specific suggestions to temper the study's broader applicability.

Response: We appreciate the overall positive and constructive comments and modified the manuscript according to the suggestions.

First, the study provides valuable insights into early-stage decomposition processes during the 12-week incubation period. However, the focus on short-term dynamics leaves a critical gap in understanding long-term carbon (C) stabilization mechanisms, such as microbial necromass accumulation or mineral-associated organic matter formation. These processes are essential for predicting the persistence of soil organic carbon and its role in climate change mitigation. I suggest addressing the following point: the 12-week incubation period is suitable for capturing early-stage decomposition but fails to account for long-term C stabilization pathways. Including a discussion on how the observed early-stage decomposition might influence long-term C stabilization, even if these processes were not directly measured.

Response: Thank you for these valuable comments with which we agree. Reviewer 1 had some similar remarks and we considerably modified the discussion about potential implications for longer term soil C dynamics based on our short-term decomposition experiment.

Second, the reliance on live roots, without accounting for pre-senescence nutrient resorption, risks misrepresenting decomposition processes in natural settings. Although a recent study reported that temperate root resorption rates (20-25%) are lower than foliar rates (55%) and there were no significant differences in nutrient resorption metrics of root between mycorrhizal types (Wang et al., 2025, *New Phytologist*, Nutrient resorption of leaves and roots coordinates with root nutrient acquisition strategies in a temperate forest), the generalizability of these values to subtropical tree species remains unclear. Furthermore, dead roots often undergo lignification during senescence, which live roots may not approximate. So, I suggest the authors should make it more explicit in the discussion section that the decomposition experiments based on live root tissue may differ from those using naturally senesced roots to further clarify whether the experimental approach may conservatively reflect in situ decomposition dynamics.

Response: Yes, this is a difficult issue when working with roots. It is often very difficult to distinguish live from dead roots, particularly in soil from the field as roots gradually lose their function with age (Eissenstat & Volder, 2004) being colonized by saprophytic fungi while still alive (Resendes et al., 2008). Therefore, the difference between living roots and decomposed roots represents a continuum, making it extremely difficult to collect dead roots that have not yet begun to decompose (Hobbie et al., 2010). Moreover, the most distal roots are very fragile, so even if the dead roots can be determined, it remains difficult to collect a sufficient amount of dead distal roots (Ma et al., 2015). With all this in mind, we had little choice than to work with live roots, acknowledging that truly senesced roots will differ in their quality and likely in their decomposition. We now added a paragraph to the discussion referring to this inherent problem when working with roots as Rev. 2 requested. Please see lines 411-421.

“A recurrent difficulty in root decomposition work is the use of live root tissues as it is

excessively difficult to extract naturally senesced root litter from the soil, especially for a large number of species as in our case. Some of the reported root trait values could therefore differ in true root litter, in particular those influenced by resorption processes such as nutrients. This would likely influence decomposition, as well as the reported trait dissimilarity among species. Our data may thus, underestimate trait dissimilarity and differences in decomposition because differences in resorption efficiency during senescence is expected to increase interspecific differences in nutrient-related traits^{42,43,44}, although potentially less so in roots than in leaves with lower resorption in roots compared to leaves⁴⁵.”

Third, while the focus on condensed tannins, nitrogen, and their ratio provides valuable insights into decomposition dynamics, a more comprehensive analysis integrating lignin content and non-structural carbohydrates would significantly enhance the mechanistic interpretation of results. Lignin, as a dominant recalcitrant compound, directly constrains microbial degradation through physicochemical barriers, while non-structural carbohydrates (e.g., soluble sugars, starch) can prime microbial activity, transiently accelerating early-stage decomposition even in tannin-rich tissues. To mitigate this, I suggest framing conclusions more cautiously and explicitly proposing future work to disentangle their roles.

Response: We also agree with this third major comment and modified the discussion accordingly with more cautious conclusions about trait control. Please see lines 405-411.

“We may have missed some additional root traits in our analysis, such as lignin, bound phenolics, non-structural carbohydrates or nutrients other than N, which add to trait divergence in roots²⁷ and may contribute to regulate decomposition¹⁵. It would be interesting to evaluate the robustness of the relative role of root traits in driving root mixture decomposition in future studies including a wider spectrum of root traits. A recurrent difficulty...”

Fourth, the manuscript specifies the use of 1000 bootstrap iterations for statistical analysis, which is a reasonable choice for balancing computational efficiency and precision. While 1000 iterations are commonly used in bootstrap analysis, the stability of the results should be explicitly verified, especially for readers who may question whether this number is sufficient for the specific dataset and analysis. Without sensitivity analysis, it is unclear whether the results would significantly change if a higher number of iterations were used. Conducting a sensitivity analysis by comparing results across different numbers of bootstrap iterations would involve recalculating key statistical metrics (e.g., means, standard deviations, confidence intervals) for each iteration count. Please include a supplementary figure or table summarizing the findings of the sensitivity analysis.

Response: Thanks for these valuable comments. We have conducted a comprehensive sensitivity analysis comparing results across different bootstrap iterations (500, 1000,

2000, 5000). We have added the key statistical metrics in the revised manuscript (lines 731-734), and the related results were presented in the new Supplementary Tables 6 and 7.

“We further conducted a sensitivity analysis of the established bootstrap method with different replacements (iteration numbers = 500, 1000, 2000, 5000), and metrics showing relative variation $((\max - \min)/\min) < 10\%$ were stable.”

Fifth, please upload complete leaf litter dataset (species, locations, climate) to Dryad/Figshare, and provide R scripts for hierarchical partitioning analyses to ensure methodological transparency and reproducibility of the results.

Response: We have uploaded the complete leaf litter dataset to the Figshare website and provided R scripts for hierarchical partitioning analyses and other analyses used in our manuscript to ensure methodological transparency and reproducibility of the results.

References:

- Eissenstat, D. M. & Volder, A. The efficiency of nutrient acquisition over the life of a root. *Nutrient Acquisition by Plants: Ecological Perspective* **181**, 185-220 (2004).
- Hobbie, S. E., Oleksyn, J., Eissenstat, D. M. & Reich, P. B. Fine root decomposition rates do not mirror those of leaf litter among temperate tree species. *Oecologia* **162**, 505-513 (2010).
- Ma, C., Xiong, Y., Li, L. & Guo, D. Root and leaf decomposition become decoupled overtime: implications for below- and above-ground relationships. *Functional Ecology* **30**, 1239-1246 (2015).
- Resendes, M. L., Bryla, D. R. & Eissenstat, D. M. Early events in the life of apple roots: variation in root growth rate is linked to mycorrhizal and nonmycorrhizal fungal colonization. *Plant and Soil* **313**, 175-186 (2008).